# Leucine-Enriched Diet Reduces Fecal MPO but Does Not Protect Against DSS Colitis in a Mouse Model of Crohn’s Disease-like Ileitis

**DOI:** 10.3390/ijms252111748

**Published:** 2024-11-01

**Authors:** Drishtant Singh, Paola Menghini, Alexander Rodriguez-Palacios, Luca Di Martino, Fabio Cominelli, Abigail Raffner Basson

**Affiliations:** 1Department of Nutrition, School of Medicine, Case Western Reserve University, Cleveland, OH 44106, USA; dxs1087@case.edu; 2Division of Gastroenterology & Liver Diseases, School of Medicine, Case Western Reserve University, Cleveland, OH 44106, USA; pxm295@case.edu (P.M.); axr503@case.edu (A.R.-P.); fxc90@case.edu (F.C.); 3Digestive Health Research Institute, University Hospitals Cleveland Medical Center, Cleveland, OH 44106, USA; 4Mouse Models Core, Silvio O’Conte Cleveland Digestive Diseases Research Core Center, Cleveland, OH 44106, USA; 5Germ-Free and Gut Microbiome Core, Digestive Health Research Institute, Case Western Reserve University, Cleveland, OH 44106, USA; 6Department of Molecular Biology and Microbiology, Case Western Reserve University, Cleveland, OH 44106, USA; 7Case Digestive Health Research Institute, School of Medicine, Case Western Reserve University, Cleveland, OH 44106, USA; lxd150@case.edu; 8Department of Medicine, School of Medicine, Case Western Reserve University, Cleveland, OH 44106, USA

**Keywords:** leucine, anti-inflammatory, diet, DSS colitis, SAMP1/YitFC mice

## Abstract

Understanding the complex link between inflammation, gut health, and dietary amino acids is becoming increasingly important in the pathophysiology of inflammatory bowel disease (IBD). This study tested the hypothesis that a leucine-rich diet could attenuate inflammation and improve gut health in a mouse model of IBD. Specifically, we investigated the effects of a leucine-rich diet on dextran sulfate sodium (DSS)-induced colitis in germ-free (GF) SAMP1/YitFC (SAMP) mice colonized with human gut microbiota (hGF-SAMP). hGF-SAMP mice were fed one of four different diets: standard mouse diet (CHOW), American diet (AD), leucine-rich AD (AD + AA), or leucine-rich CHOW diet (CH + AA). Body weight, myeloperoxidase (MPO) activity, gut permeability, colonoscopy scores, and histological analysis were measured. Mice on a leucine-rich CHOW diet showed a decrease in fecal MPO prior to DSS treatment as compared to those on a regular diet (*p* > 0.05); however, after week five, prior to DSS, this effect had diminished. Following DSS treatment, there was no significant difference in gut permeability, fecal MPO activity, or body weight changes between the leucine-supplemented and control groups. These findings suggest that while a leucine-rich diet may transiently affect fecal MPO levels in hGF-SAMP mice, it does not confer protection against DSS-induced colitis symptoms or mitigate inflammation in the long term.

## 1. Introduction

Dietary amino acids are critical regulators of cellular and microbial metabolic pathways and play crucial functions in gut homeostasis [1]. Specifically, in inflammatory bowel disease (IBD), amino acids have been shown to play an important role in gut inflammation and modulation of gut microbiota with amino acid metabolism linked to IBD severity [2,3,4,5,6].

A growing number of chronic inflammatory diseases and metabolic conditions are attributed to various environmental factors or modern diets, such as the Western diet, which is characterized by high animal protein, saturated fats, and refined carbohydrates [7,8]. In terms of dietary protein, both a lack or an abundance of dietary amino acids (i.e., amino acid-deficient or amino acid-enriched diet) has also been shown to worsen the severity of colitis [5,9], as well as reduce inflammation and oxidative stress in experimental IBD [10,11,12,13]. However, investigating the relationship between diet and chronic gastrointestinal disorders in humans is challenging.

Amino acids are essential building blocks for metabolically active proteins, glutathione (GSH), nitric oxide, polyamines, and other compounds in the intestinal mucosal cells, thus, helping in promoting intestinal growth and maintaining the mucosal integrity and barrier function [14]. Furthermore, amino acids aid in the production of macromolecules required for intestinal mucosal wound repair and serve as an energy source for enterocytes. The gut microbiota can use these amino acids to build proteins and other metabolites, impacting the host’s nutrition and function [15]. Some amino acids including glutamine and arginine have been shown to influence the course of IBD by potentially reducing inflammation, oxidative stress, and proinflammatory cytokine levels [16,17]. Recently, our lab identified a negative correlation between fecal leucine and intestinal inflammation [18]. The potential role of dietary leucine is further supported by human studies showing decreased leucine concentrations during active disease [19,20]. The discrepancies in serum amino acid levels between IBD patients and healthy people, as well as the incidence of malnutrition in IBD, underline the need for innovative adjunct medications that address these underlying metabolic imbalances [21,22,23].

The study of human IBD prototypes, specifically CD and UC, has been significantly advanced through the development of the SAMP mouse model, which is characterized by a 100% incidence of transmural inflammation and cobblestone-like lesions in the ileum [24]. By the age of 14 weeks, these mice exhibit pathological features strikingly similar to those seen in human CD. To explore dietary effects on intestinal inflammation, particularly in UC, the dextran sodium sulfate (DSS) model is the most widely adopted protocol across laboratories studying IBD/UC (28560286). We specifically focused on the SAMP mouse model because no other available animal models replicate the CD or induced colitis.

Herein, we investigated the effect of a leucine-rich diet, in the context of an ‘American’ diet [18] and a standard mouse laboratory diet, using germ-free SAMP1/YitFC (SAMP) mice colonized with human gut microbiota and treated with dextran sulfate sodium (DSS). We hypothesized that leucine-rich diets would be anti-inflammatory and provide protection against DSS-induced colitis.

## 2. Results

### 2.1. Leucine-Rich Diet Had a Short-Term Effect of Fecal MPO in hGF-SAMP Mice

To determine the effect of dietary leucine in hGF-SAMP mice, mice were fed one of four diets for 6 weeks and then treated with DSS. As an in vivo measure of inflammation, we measured fecal MPO each week. Prior to DSS treatment, mice fed the leucine-rich diet exhibited a significantly lower fecal MPO compared to those fed the standard diet (fecal MPO activity; CH + AA: 3.0 ± 0.8 vs. CH: 4.9 ± 0.7; *p* = 0.003, week 5); however, this effect disappeared by week 6. The MPO activity was decreased in the leucine-rich diet groups temporarily for up to 5 weeks as compared to the control groups (Figure 1A–G). After 6 weeks of diet administration, the % residual weight of leucine-rich diet groups was lower than that of control groups (Figure 1H).

### 2.2. Mice Fed Dietary Leucine Have Increased DSS-Induced Weight Loss Compared to Controls

Following 6 weeks of diet, mice groups were treated with 3% DSS for 7 days to induce acute colitis. Mice fed leucine-rich diets exhibited comparable colitis symptoms (body weight, colonoscopy, histology, fecal MPO, and gut permeability) compared to their DSS-treated controls. There was no significant difference in the % change in body weight of the mice fed with leucine-rich AD compared to the control group (% residual weight on the day of sacrifice; AD + AA: 72.4 ± 6.8, AD: 70.5 ± 6.5, ANOVA, *p* = 0.06) (Figure 2A). However, the body weight of mice fed with a leucine-rich CHOW diet was significantly lower than that of the control group (CH + AA: 82.7 ± 14.0, CH: 98.4 ± 3.8, ANOVA, *p* = 0.06). There was no significant difference in the fecal MPO activity between the leucine-rich diet groups and the control groups after DSS (fecal MPO activity log_2_; AD: 5.3 ± 2.0, AD + AA: 4.6 ± 1.5, CH: 3.7 ± 1.8, CH + AA: 5.6 ± 0.5; ANOVA, *p* = 0.3) (Figure 2B,C). There was also no improvement in gut permeability as determined by the translocation of FITC–dextran from the gut lumen into the plasma (FITC–dextran log_2_; AD + AA: 21.4 ± 0.9, AD: 20.8 ± 1.3, CH + AA: 20.1 ± 1.5, CH: 18.7 ± 0.7; ANOVA, *p* = 0.005) (Figure 2D,E). No significant difference was identified in post-DSS colonoscopy and histology scores between the leucine-rich diet groups and controls. The data presented in Appendix A represent.

## 3. Discussion

Leucine is an aliphatic amino acid found in abundance in protein-rich foods [25]. The aliphatic amino acids, including leucine, isoleucine, and valine, have received considerable interest for their functions that extend beyond conventional metabolic processes [26,27,28]. Despite the well-known link between elevated circulating levels of various amino acids and their role in various diseases [29,30,31,32], limited research has been conducted on the dietary intake of leucine in the context of IBD. In this study, we assessed the effect of a diet rich in leucine on DSS-induced colitis in hGF-SAMP mice. Our findings suggest that added dietary leucine significantly decreased intestinal inflammation for up to four weeks, as evident from the fecal MPO activity; however, leucine had no significant effect on DSS-colitis severity and intestinal permeability. Of note, however, leucine-fed mice had greater DSS-induced weight loss compared to control mice.

The weekly assessment of fecal MPO activity suggested that leucine had a short-term anti-inflammatory effect on treated mice. According to previous studies conducted using amino acid diets, short-term treatment of various amino acids such as glutamine and arginine was found to help in the reduction of intestinal inflammation in rats and mice [33,34,35,36]. This indicates that the dietary intervention of amino acids over a brief period has the potential to significantly reduce inflammation.

There was no difference in post-DSS colonoscopy scores or intestinal permeability between the leucine-treated and control groups, suggesting that a diet high in leucine does not worsen or alleviate the severity of colitis. Various studies have shown that dietary amino acids may play a role in increasing or alleviating the inflammation and symptoms of colitis in animals. In one study, a diet rich in leucine prevented the invasion of inflammatory cells and accelerated the expression of proinflammatory cytokines in rats [37]. In a previous study conducted in rats, glutamine helped to protect them from colitis and decreased the inflammation and severity of colitis [38]. Similarly, dietary intervention of several other amino acids such as arginine, histidine, glycine, etc., protected from DSS-induced colitis and helped in alleviating the inflammation and related symptoms [23,39,40]. In a previous study conducted in mice, a diet rich in amino acids increased inflammation in the gut and worsened DSS-induced colitis [41].

Leucine and various other aliphatic amino acids play a key role in regulating the immune system by influencing several pathways related to cell growth, proliferation, differentiation of immune cells, production of inflammatory cytokines, and the maintenance of immune homeostasis [42,43]. Various studies have reported higher plasma concentrations of aliphatic amino acids in different inflammatory-related diseases showing a negative association between the amino acids and various diseases [44]. It was previously reported that a leucine-rich diet inhibited the infiltration of inflammatory cells and increased the expression of proinflammatory cytokines [21]. These findings suggest that leucine may play a dual role in modulating immune responses. Previous studies have also reported that the aliphatic amino acids have immune-modulating features and are linked to increased inflammation and toxic health effects in various immunological disorders [45,46]. It was previously reported in a study that a diet either low or deficient in leucine may inhibit DSS-induced colitis and reduce intestinal inflammation [9]. Leucine and other amino acids have been proven to have mixed effects on colitis, with some studies indicating protective benefits and others indicating increased inflammation. Overall, leucine might play a dual role in immune regulation, possibly regulating both inflammation and immunological responses based on nutritional and disease conditions. In conclusion, this study gives important insights into the various effects of leucine-rich diets on DSS-induced colitis in hGF-SAMP mice. Further investigation is warranted to elucidate the potential correlation between inflammation, body weight regulation, and leucine-rich diets in the context of IBD.

## 4. Materials and Methods

### 4.1. Animals

This experiment tested groups of (6 mice/group) of age- and sex-matched 14-week-old germ-free (GF) SAMP1/YitFc (SAMP) mice. The SAMP GF mouse colony (Cleveland Digestive Diseases Research Core Center—CDDRCC, Mouse Models Core) is maintained in high-efficiency particulate air-filtered pressurized isolators in the Animal Resource Center ultra-barrier facility, at the Case Western Reserve University (CWRU) School of Medicine. All experiments were conducted in BSL-2 grade rooms with dedicated use for gnotobiotic animals.

All mice were caged using our GF-grade nested isolation (NesTiso) caging system and maintained on non-edible Aspen bedding provided by the CDDRCC Mouse Models Core. Mice were subjected to a 12 h light and 12 h dark cycle in AAALAC-accredited Animal Research Center rooms at Case Western Reserve University, in species-appropriate temperature and humidity-controlled rooms. Protocols on animal handling, housing, and the transplant of human microbiota into GF mice were approved by the IACUC and the Institutional Review Board at CWRU, following the National Research Council Guide for the Care and Use of Laboratory Animals (2014–0158). Measures to control for bedding-dependent microbial bias/overgrowth were implemented in all experiments, as previously described [47,48].

### 4.2. Transplantation of Human Gut Microbiota

The establishment of human gut microbiota in GF mice was performed using previously characterized cryopreserved human fecal microbiota communities isolated from a patient with Crohn’s disease (stored at −80 °C; phosphate-buffered saline (PBS)/7% dimethyl sulfoxide (DMSO) mixture), as previously described [47]. All GF mice were gavaged with gut microbiota (0.20 mL/10 g of body weight; 10^8–9^ colony-forming units; CFU/mouse) two weeks prior to starting the experiment.

### 4.3. Diet and Experimental Design

Mice were randomized to one of four dietary groups: the American diet (AD), the leucine-rich AD (AD + AA), a standard mouse laboratory diet P3000 of CHOW (CH), or the leucine-rich CHOW diet (CH + AA) (Figure 3). The AD used in this study was originally created to mirror the 2011–2012 National Health and Nutrition Examination Survey’s (NHANES) “What We Eat in America” survey and has been described in detail by our laboratory [18]. The modifications in the diets were carried out by Research Diets, NJ and the composition of all four diets is described in the Appendix A. The leucine content of the CH + AA diet was ~14.3 mg/g in the diet and the daily intake of leucine received by mice in this group was around ~72 mg (assuming ∼5 g daily intake of diet). However, the leucine content of the CH + AA diet was ~28.1 mg/g in the diet and the daily intake of leucine received by mice in this group was around ~141 mg (assuming ∼5 g daily intake of diet). Body weight, myeloperoxidase (MPO) activity, and ileitis severity (colonoscopy, histology) were measured in mice. Colitis was induced using dextran sodium sulfate (DSS) after six weeks on the various diets. At the end of the experiment, mice were humanely euthanized using carbon dioxide narcosis. All tests were carried out in accordance with criteria aimed at decreasing microbiome variability and enhancing the study’s reproducibility and statistical power [49,50].

### 4.4. DSS-Induced Colitis

To induce colitis, all mice were given ad libidum 3% (weight/volume) DSS dissolved in sterile water for 7 days. Mice then resumed with regular water for two days and then were sacrificed. To assess the impact of colitis, the mice were observed daily for changes in body weight, as well as the occurrence of blood, the consistency of their feces, and their general appearance.

### 4.5. Colonoscopy

Following DSS, colonoscopic assessment was used to measure inflammation in the colon using a previously validated scoring method [51]. In brief, isoflurane, USP (Butler Schein Animal Health), was used to anesthetize mice prior to endoscopic operations, and no laxatives or fasting was required to prepare for the colonoscopy. After that, the mice were euthanized, and numerous parameters such as BW and colitis severity were measured.

### 4.6. Analysis of Intestinal Inflammation and Gut Permeability In Vivo

MPO activity in feces was measured weekly, and intestinal gut permeability (fluorescein isothiocyanate; FITC–dextran) was assessed before and after DSS treatment. Fecal MPO activity was determined using a previously established dianisidine-H_2_O_2_ technique adapted for use with 96-well plates [52]. A previously established method was used to assess intestinal gut permeability in mice [53]. In brief, the mice were fasted overnight before receiving an oral gavage of a fluorescently labeled sugar probe, FITC–dextran (80 mg/mL in sterile PBS). After 4 h, blood was collected in EDTA-coated vials and plasma was diluted properly to measure the fluorescence intensity, which correlates with intestinal permeability.

### 4.7. Histopathological Analysis

Colons were rinsed with sterile phosphate-buffered saline. Following that, a longitudinal incision was made in the colon and the diseased tissue was preserved in 10% buffered formalin. Following a 24 h fixative stage at 4 °C, the specimens underwent a 70% ethanol washing process. Tissues were then paraffin-embedded and subjected to staining with Hematoxylin and Eosin (H&E). Intestinal inflammation was then scored in a blinded manner using a validated scoring system [24,54]. The scoring system assessed the following three parameters: (1) active inflammation, (2) chronic inflammation, and (3) villus architecture. A numerical score was allocated to each of these factors, with values ranging from zero (indicating normal tissue) to three (indicating increasing levels of inflammation).

## Figures and Tables

**Figure 1 ijms-25-11748-f001:**
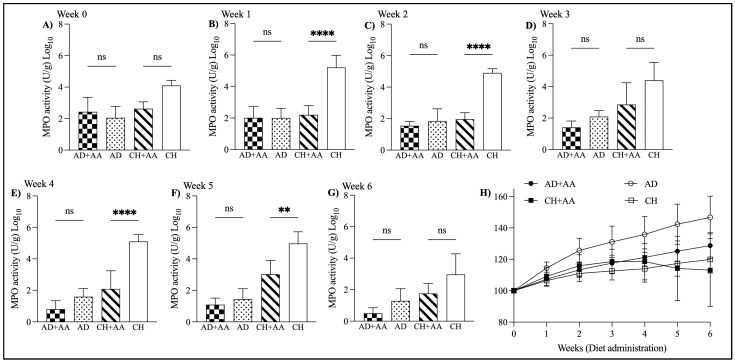
MPO activity in mice over 6 weeks. (**A**–**G**) MPO activity after 1, 2, 3, 4, 5, and 6 weeks of diet administration. Note, there was no significant difference between groups in fecal MPO at baseline, prior to starting the diets. (**H**) Percentage change from original body weight over 6 weeks (defined as day 0 and as 100%) (ns *p* ≥ 0.05, ** *p* < 0.01, **** *p* < 0.0001).

**Figure 2 ijms-25-11748-f002:**
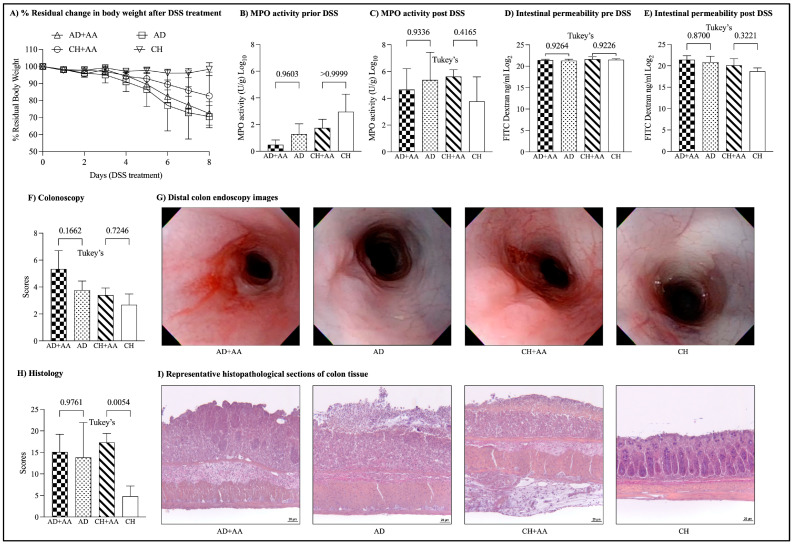
Leucine supplementation does not reduce severity of acute chemical colitis. (**A**) Percentage change from original body weight (defined as day 0 and as 100%) after induction of DSS-colitis, (**B**) MPO activity before and (**C**) after DSS treatment, (**D**) intestinal permeability assay (FITC–dextran) before and (**E**) after DSS treatment, (**F**) colonoscopy score, (**G**) distal colon endoscopy images, (**H**) histology score, (**I**) representative histopathological sections of colon tissue (DSS, dextran sulfate sodium; FITC, fluorescein isothiocyanate; MPO, myeloperoxidase).

**Figure 3 ijms-25-11748-f003:**
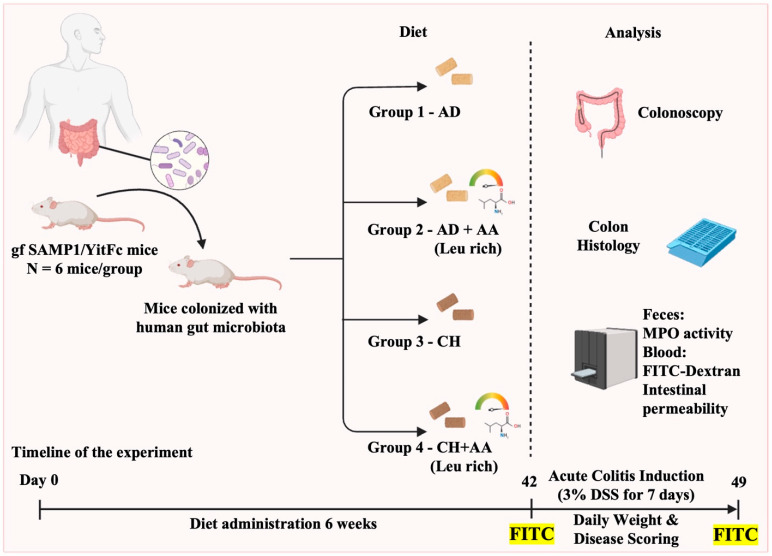
Experimental design. (gf, germ free; SAMP1/YitFc, a sub-strain of AKR/J mice produced through a program of selective breeding; AD, American diet; AD + AA, leucine-rich American diet; CH, standard CHOW diet; CH + AA, leucine-rich CHOW diet; Leu, leucine; DSS, dextran sulfate sodium; FITC, fluorescein isothiocyanate; MPO, myeloperoxidase.).

## Data Availability

The original contributions presented in this study are included in the article/Appendix A, further inquiries can be directed to the corresponding author.

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
