# Peer review of "Leucine-Enriched Diet Reduces Fecal MPO but Does Not Protect Against DSS Colitis in a Mouse Model of Crohn’s Disease-like Ileitis"

_ijms, 2024, doi:10.3390/ijms252111748_

Round 1
Reviewer 1 Report
Comments and Suggestions for Authors
The study is interesting and has merit. However, several important points must be clarified.
1. The abstract must be better focused and present a clear hypothesis for the study.
2. The methods are clear, however, it is important to present a NAIVE control group.
3. It is not clear how much leucine was received by the animals.
4. The authors present a supplementary table with several things. Was each diet made up of all those ingredients?
5. What is the clear composition of the AD diet and why was it chosen?
6. Are the MPO values ​​of the CH group increased or are they the same as normal animals? This information is not clear.
7. What is the real control group?
8. What was the food consumption and body weight of the animals like over the 6 weeks? This data must be entered.
9. What is the explanation for MPO being increased in CH animals?
10. How were the inflammatory cytokine values ​​in the groups?
11. How to explain the lower MPO values ​​in the CH group, in fig. 2B compared to CH in Fig. 1 (Week5). Does induction with DSS decrease MPO?
12. The colonoscopy images must be presented and not just a graph with scores.
13. Histological images of all groups must be added. In addition, a table with the appropriate scores for inflammation, infiltration, etc...
14. The discussion is very short and not very informative. It must be deepened with a focus on the study and discussing the mechanisms involved.
Author Response
Comment 1. The abstract must be better focused and present a clear hypothesis for the study.
Response: We have corrected the abstract according to the suggestions by the reviewer.
Comment 2. The methods are clear, however, it is important to present a NAIVE control group.
Response: The mice on the non-modified standard CHOW diet i.e. P3000 (CH group) are considered as the NAIVE control group.
Comment 3. It is not clear how much leucine was received by the animals.
Response: The leucine content of CH+AA diet was ~14.3 mg/g in the diet and the daily intake of leucine received by mice in this groups was around ~ 72 mg (assuming ∼5gdaily intake of diet). However, the leucine content of CH+AA diet was ~28.1 mg/g in the diet and the daily intake of leucine received by mice in this groups was around ~ 141 mg (assuming ∼5gdaily intake of diet). The same has been added to the MS text. (Line 219-224)
Comment 4. The authors present a supplementary table with several things. Was each diet made up of all those ingredients?
Response: Each diet is composed of all the ingredients present in the supplementary table.
Comment 5. What is the clear composition of the AD diet and why was it chosen?
Response: The composition of the AD diet was formulated based on the composition of the 2011–2012 cycle of “What We Eat in America” by NHANES (National Health and Nutrition Examination Survey). The NHANES diet mirrors the classical Western Diet. In our studies, the AD was chosen as it is high in animal protein, refined carbohydrates and is believed to significantly contribute to the increasing rates of inflammatory bowel diseases like Crohn's disease and ulcerative colitis (PMID 28804483, 31072001).
Comment 6. Are the MPO values ​​of the CH group increased or are they the same as normal animals? This information is not clear.
Response: The MPO values of CH group were higher than the other groups. At the start of the experiment, there is no significant difference in the MPO values between groups. However, after each week the MPO values for all groups decreased (except for CH that only decreased to some extent after week 6). Whereas, after DSS treatment MPO values of CH group was lower than that of the groups (Section 2.1).
Comment 7. What is the real control group?
Response: Each diet group; either AD or CH i.e. without leucine serves as control. But the CH serves as the overall NAIVE control.
Comment 8. What was the food consumption and body weight of the animals like over the 6 weeks? This data must be entered.
Response: The food was replaced every week and due to hygroscopic nature of the research diets, the food consumption could not be calculated. However, we have provided the body weight of animals in each group in the paper.
Comment 9. What is the explanation for MPO being increased in CH animals?
Response: The MPO values of CH animals were not increased. At the start of the experiment, there is no significant difference in the MPO values between groups. However, after each week the MPO values for all groups decreased (except for CH that only decreased to some extent after week 6).
Comment 10. How were the inflammatory cytokine values ​​in the groups?
Response: The inflammatory cytokine profile was not evaluated in this study.
Comment 11. How to explain the lower MPO values ​​in the CH group, in fig. 2B compared to CH in Fig. 1 (Week5). Does induction with DSS decrease MPO?
Response: The change in MPO values in CH group before and after DSS is not statistically significant. DSS induction does not directly decrease MPO levels in this context. The variation in MPO might be technical variation as both the values are from assays performed on different days.
Comment 12. The colonoscopy images must be presented and not just a graph with scores.
Response: The colonoscopy images has been added to the MS as suggested by the reviewer.
Comment 13. Histological images of all groups must be added. In addition, a table with the appropriate scores for inflammation, infiltration, etc...
Response: The histological images and the appropriate table have been added to the MS as per the reviewer’ suggestions.
Comment 14. The discussion is very short and not very informative. It must be deepened with a focus on the study and discussing the mechanisms involved.
Response: The discussion part has been revised according to the suggestions by reviewer.

Reviewer 2 Report
Comments and Suggestions for Authors
The authors of this study examined the impact of a leucine-rich diet (LRD) on DSS-induced colitis in germ-free SAMP1/YitFC (SAMP) mice colonized with human gut microbiota (hGF-SAMP).They examined four diets on mice: the regular (chow) diet, the "American" diet, the leucine-rich "American" diet, and the leucine-rich chow diet.Researchers found that LRD had a short-term effect on the levels of MPO in the feces of hGF-SAMP mice. Animals that were fed LRD lost more weight than animals that were fed controls.
The use of English is perfectly adequate.
The descriptions of the methods are correct. The method used is adequate. The references are also adequate.
However, several aspects need improvement.
My overall impression of the experiment is that its goal is exciting and its structure is good, but the authors' objective is difficult to follow in terms of execution and results presentation.
The Introduction chapter should highlight that SAMP mice spontaneously develop Crohn's-like ileitis, while DSS colitis primarily acts as an animal model of ulcerative colitis, mainly inducing inflammation in the distal colon.
They also need to explain why they combined Crohn's and ulcerative colitis (using SAMP mice for DSS induction).
They also need to explain why they only studied ileitis in animals with DSS-colitis induction.
The results of the colon examination remain unclear. If they killed the animals and processed the ileum and colon, why was a colonoscopy necessary?
After slaughter, could no mucosal abnormalities in the ileum be macroscopically detected?
The text in subsection 2.1 should clearly state that LRD on a chow diet caused a temporary decrease in MPO activity.
Asterisks should indicate significant differences in Figure 1 for ease of interpretation.
The discussion section essentially reiterates all the findings, but it provides no molecular or immunobiological explanation for why this might have happened in the experimental animals.
I strongly recommend the inclusion of representative histological photographs of the ileum and colon.
Based on the above, I recommend a major revision.
Author Response
Overall Statement: The authors of this study examined the impact of a leucine-rich diet (LRD) on DSS-induced colitis in germ-free SAMP1/YitFC (SAMP) mice colonized with human gut microbiota (hGF-SAMP).They examined four diets on mice: the regular (chow) diet, the "American" diet, the leucine-rich "American" diet, and the leucine-rich chow diet.Researchers found that LRD had a short-term effect on the levels of MPO in the feces of hGF-SAMP mice. Animals that were fed LRD lost more weight than animals that were fed controls. The use of English is perfectly adequate. The descriptions of the methods are correct. The method used is adequate. The references are also adequate. However, several aspects need improvement.
Response: Thank you for the comments and suggestions.
Specific comments:
Comment 1. My overall impression of the experiment is that its goal is exciting and its structure is good, but the authors' objective is difficult to follow in terms of execution and results presentation.
Response: We have revised the manuscript according to the reviewer’s comments.
Comment 2. The Introduction chapter should highlight that SAMP mice spontaneously develop Crohn's-like ileitis, while DSS colitis primarily acts as an animal model of ulcerative colitis, mainly inducing inflammation in the distal colon.
Response: We have added the explanation for SAMP model in the introduction part as suggested by the reviewer.
Comment 3. They also need to explain why they combined Crohn's and ulcerative colitis (using SAMP mice for DSS induction).
Response: We focused on SAMP mice because no other model lines exist to resemble CD-like cobblestone ileitis
Comment 4. They also need to explain why they only studied ileitis in animals with DSS-colitis induction.
Response: The dextran sodium sulfate (DSS) mouse model is the most commonly used for colitis research. DSS, a chemical colitogen with anticoagulant properties, induces epithelial damage, making the model popular for its speed, simplicity, reproducibility, and control in IBD studies.
Comment 5. The results of the colon examination remain unclear. If they killed the animals and processed the ileum and colon, why was a colonoscopy necessary?
Response: The colonoscopy was performed prior to sacrifice of the animals. This technique provides visual evidence of mucosal damage, ulceration, and colonic inflammation, which helps to monitor the severity of colitis in live animals.
Comment 6. After slaughter, could no mucosal abnormalities in the ileum be macroscopically detected?
Response: As the study was focused on DSS induced colitis that represents UC, only colon tissue was examined for the histopathological features.
Comment 7. The text in subsection 2.1 should clearly state that LRD on a chow diet caused a temporary decrease in MPO activity.
Response: The text was modified as suggested by the reviewer.
Comment 8. Asterisks should indicate significant differences in Figure 1 for ease of interpretation.
Response: Asterisks have been added on all panels in Figure 1.
Comment 9. The discussion section essentially reiterates all the findings, but it provides no molecular or immunobiological explanation for why this might have happened in the experimental animals.
Response: We have revised the discussion part as suggested by the reviewer.
Comment 10. I strongly recommend the inclusion of representative histological photographs of the ileum and colon.
Response: The histological images have been added to the MS as per the reviewer’s suggestions.
Round 2
Reviewer 2 Report
Comments and Suggestions for Authors
The authors corectly revised the manuscript.